## Low-density lipoprotein cholesterol and all-cause mortality: findings from the China health and retirement longitudinal study

Liang Zhou,[1] Ying Wu,[2] Shaobo Yu,[3] Yueping Shen,[4] Chaofu Ke [ID] [4]

LZ and YW contributed equally.

For numbered affiliations see end of article.

**Correspondence to**
Dr Chaofu Ke;
cfke@suda.edu.cn and
Dr Yueping Shen;
shenyueping@suda.edu.cn

## ABSTRACT

**Objectives** To investigate the relationship between low-density lipoprotein cholesterol (LDL-C) and all-cause mortality among middle-aged and elderly Chinese population.

**Design** Prospective cohort study.

**Setting** This study used data from the China Health and Retirement Longitudinal Study.

**Participants** Middle-aged and elderly participants with complete data were enrolled for a 4-year follow-up of total mortality and plasma levels of LDL-C, including 4981 male respondents and 5529 female respondents.

**Results** During a 4-year follow-up, there were 305 and 219 deaths in men and women, respectively. Compared with the first quintile (Q1) of LDL-C, the adjusted HRs (95% CIs) were 0.818 (0.531 to 1.260) for Q2, 0.782 (0.507 to 1.208) for Q3, 0.605 (0.381 to 0.962) for Q4 and 0.803 (0.506 to 1.274) for Q5 in men. The results from restricted cubic spine (RCS) showed that when the 20th percentile of LDL-C levels (84 mg/dL) was used as the reference, a lower LDL-C concentration (<84 mg/dL) was associated with a higher 4-year all-cause mortality risk. By contrast, both quintile analysis and RCS analysis did not show a statistically significant association in women.

**Conclusions** Compared with moderately elevated LDL-C (eg, 117–137 mg/dL), a lower plasma level of LDL-C (eg, ≤84 mg/dL) was associated with an increased risk of 4-year all-cause mortality in middle-aged and elderly Chinese men. The results suggest the potential harmful effect of a quite low level of LDL-C on total mortality.

## INTRODUCTION

For decades, the mainstream view holds that a high level of low-density lipoprotein cholesterol (LDL-C) is a primary cause of cardiovascular events and mortality.[1] However, many studies have found that LDL-C levels are inversely associated with all-cause mortality in diseased populations, such as patients with chronic haemodialysis,[2] intracerebral haemorrhage[3] and heart failure.[4] Most importantly, in a systematic review of 19 cohort studies including 30 cohorts with 68 094 elderly people (≥60 years), an inverse association between LDL-C and all-cause mortality was

### Strengths and limitations of this study

► This study used high-quality data from a nationally representative longitudinal cohort to investigate the relationship between low-density lipoprotein cholesterol (LDL-C) and all-cause mortality among middle-aged and elderly Chinese population.

► The use of restricted cubic spline provided a more comprehensive spectrum of the non-linear relation between LDL-C and all-cause mortality.

► Although the public's attention focuses on the benefit of lipid lowering, this study highlights the potential harmful effect of very low LDL-C.

► The 4-year follow-up period prevented the assessment of a long-term association between LDL-C and all-cause mortality.

► The unavailability of cause-specific mortality data prevented the analysis of the association between LDL-C and cause-specific mortality.

seen in 16 cohorts representing 92% of all participants, and none of the other cohorts found the positive association between LDL-C and all-cause mortality.[5] Since LDL-C has been regarded as 'bad cholesterol' for a long time and lipid-lowering drugs have been prescribed even at normal levels of serum cholesterol,[6] the impact can be substantial. Therefore, the underlying relationship between LDL-C and all-cause mortality needs to be clarified in large prospective cohorts.

In this study, we aimed to investigate whether LDL-C levels are associated with all-cause mortality among middle-aged and elderly Chinese men and women, based on the longitudinal data from the China Health and Retirement Longitudinal Study (CHARLS).

## METHODS
### Study design

As a nationally representative longitudinal study, CHARLS is designed to collect a wide

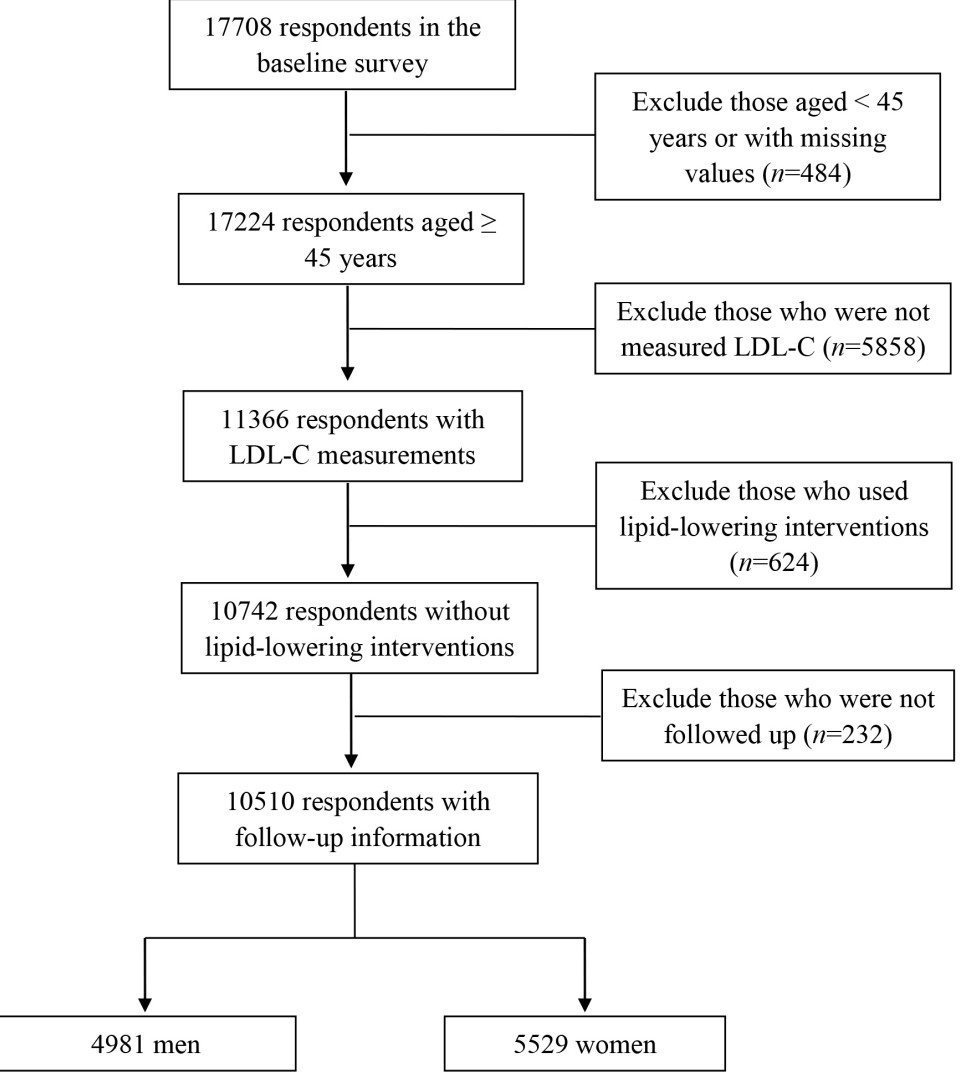

**Figure 1** Flow chart on the selection of eligible participants. LDL-C, low-density lipoprotein cholesterol.

range of information on the economic standing, physical and psychological health, demographics and social networks of a middle-aged and elderly Chinese population (aged ≥45 years).[7] The national baseline survey (wave 1) was conducted between June 2011 and March 2012 and included 17 708 respondents. The second wave (wave 2) was carried out in 2013–2014, the third wave (wave 3) in 2014–2015 and the fourth wave (wave 4) in 2015–2016. The detailed design of CHARLS can be referred to a previous publication.[7] All participants signed informed consents.

**Study population**

All participants recruited in the national baseline survey were included if they met the following criteria: (1) aged ≥45 years, (2) measured plasma levels of LDL-C in wave 1, (3) successfully followed up in at least one of the subsequent three waves and (4) without lipid-lowering interventions. Finally, 10 510 participants, including 4981 men and 5529 women, were included for subsequent analysis (figure 1).

**Plasma LDL-C measurements and other covariates**

Plasma samples were collected by medically trained staff and then stored at −80°C until assayed at Capital Medical University laboratory. LDL-C was measured by the enzymatic colorimetric test, with an analytical range of 3–400 mg/L and between-assay coefficient of variation of 1.20%. During the testing of the CHARLS study samples, quality control (QC) samples were used daily. All test results from QC samples were within 2 SD of mean QC control concentrations. The other covariates collected included age, gender, smoking status, drinking status, body mass index (BMI), educational level, household income, living alone status, rural residence, activity of daily living (ADL) disability, high-density lipoprotein cholesterol (HDL-C), triglyceride, haemoglobin, hypertension (defined by a history of hypertension, or systolic blood pressure (SBP) ≥140 mm Hg, or diastolic blood pressure (DBP) ≥90 mm Hg), high blood sugar (HBS)/diabetes (defined by a history of HBS/diabetes, or fasting blood glucose ≥6.1 mmol/L, or non-fasting blood glucose ≥7.8 mmol/L),

a history of cancer, heart disease, stroke, asthma, lung disease, liver disease, digestive disease, kidney disease, arthritis, memory problem and psychological problem. ADL covers the following items: dressing, bathing and showering, eating, getting in/out bed, using the toilet and controlling urination or defecation. Every item in the ADL scale has a four-scale answer for each question: 'no difficulty', 'have difficulty but can still do it', 'have difficulty and need help', and 'can not do it'. ADL was assigned a value of 0 if the respondents had no difficulty in all these activities and one otherwise. Hand grip strength was measured with a dynamometer (Yuejian WL-1000, Nantong, China) in kilograms twice on each hand. The mean score of two measures in the dominant hand was calculated to define hand grip strength in this study.

**Table 1** Characteristics of the study population

| Characteristics | Men (n=4981) | Women (n=5529) | P value |
|---|---|---|---|
| Age-year | 59 (53–66) | 57 (51–65) | <0.0001 |
| BMI-kg/m$^2$ | 22.40 (20.35–24.83) | 23.51 (21.17–26.14) | <0.0001 |
| SBP-mm Hg | 127.67 (115.67–141.67) | 127.00 (114.00–143.33) | 0.3764 |
| DBP-mm Hg | 75.33 (67.67–83.67) | 74.33 (67.00–82.67) | 0.0017 |
| Hand grip strength (kg) | 36.85 (30.50–43.00) | 25.00 (20.00–29.50) | <0.0001 |
| Above-average household income-no (%) | 2104 (49.68) | 2360 (50.32) | 0.5468 |
| Education level-no (%) | | | <0.0001 |
| 1 | 4287 (86.07) | 5134 (92.86) | |
| 2 | 592 (11.89) | 347 (6.28) | |
| 3 | 102 (2.05) | 48 (0.87) | |
| ADL disability-no (%) | 716 (14.55) | 1011 (18.57) | <0.0001 |
| Living alone-no (%) | 471 (9.46) | 832 (15.05) | <0.0001 |
| Rural residence-no (%) | 3282 (65.89) | 3524 (63.74) | 0.0210 |
| Lifestyle-no (%) | | | |
| Smoking ever | 3738 (75.24) | 420 (7.63) | <0.0001 |
| Drinking ever | 3297 (66.43) | 826 (15.01) | <0.0001 |
| Disease history-no (%) | | | |
| Hypertension | 1904 (43.35) | 2245 (45.34) | 0.0530 |
| HBS/diabetes | 1460 (30.16) | 1564 (29.19) | 0.2846 |
| Cancer | 40 (0.81) | 67 (1.22) | 0.0372 |
| Stroke | 133 (2.69) | 136 (2.48) | 0.4958 |
| Heart disease | 499 (10.10) | 738 (13.47) | <0.0001 |
| Lung disease | 620 (12.54) | 508 (9.26) | <0.0001 |
| Arthritis | 1548 (31.23) | 2243 (40.80) | <0.0001 |
| Liver disease | 197 (4.00) | 188 (3.44) | 0.1326 |
| Kidney disease | 309 (6.27) | 315 (5.75) | 0.2701 |
| Digestive disease | 1030 (20.79) | 1393 (25.37) | <0.0001 |
| Asthma | 283 (5.72) | 216 (3.93) | <0.0001 |
| Psychological problem | 60 (1.21) | 94 (1.72) | 0.0336 |
| Memory problem | 97 (1.96) | 86 (1.57) | 0.1238 |
| Laboratory measurements | | | |
| LDL cholesterol-mg/dL | 109.41 (88.92–131.06) | 117.91 (96.26–141.50) | <0.0001 |
| Triglyceride-mg/dL | 96.46 (69.03–145.14) | 110.63 (79.65–159.30) | <0.0001 |
| HDL cholesterol-mg/dL | 48.71 (39.43–59.54) | 50.64 (41.75–60.31) | <0.0001 |
| Haemoglobin-(g/L) | 151 (140–162) | 136 (125–146) | <0.0001 |

Educational level: (1) Less than lower secondary education; (2) Upper secondary and vocational training; (3) Tertiary education.
ADL, activity of daily living; BMI, body mass index; DBP, diastolicblood pressure; HBS, high blood sugar; HDL, high-density lipoprotein; LDL, low-density lipoprotein; SBP, systolic blood pressure.

## All-cause mortality follow-up

Participants enrolled in wave 1 were followed up in subsequent three waves. In wave 2, both the interview status (dead or alive) and death time were recorded. In waves 3 and 4, only the interview status was recorded. For those who had the exact time on all-cause death in wave 2, the survival time was defined as the interval between the interview time of wave 1 and the death time. If the exact death time was not available in waves 3 and 4, the survival time was computed as the median of the interval between wave 1 and the specific wave with death information. For those who did not die during the follow-up period, the survival time was defined as the interval between wave 1 and the last interview wave with follow-up information.

## Patient and public involvement

Anonymised participant data were used in this study. Patients and the public were not involved in the design or conduct, or reporting, or dissemination plans of the study.

## Statistical analysis

Data were presented as median ($P_{25} \sim P_{75}$) for continuous variables and frequency (percentage) for categorical variables. Baseline characteristics between or among groups were compared by the Wilcoxon rank sum test or Kruskal-Wallis rank sum test for continuous variables and by the $X^2$ test for categorical variables. The Cox proportional HR model was used to estimate the HRs and 95% CIs of LDL-C quintiles. In addition, the association between all-cause mortality and LDL-C on a continuous scale was further examined using restricted cubic splines (RCS) incorporated in Cox proportional hazards models. Bayesian information criterion was used to determine the optimal number of knots in RCS. In this study, three knots were used in all RCS analyses, with knot locations at the 10th, 50th and 90th percentiles of LDL-C. To be consistent with quintile analyses, the reference point was the 20th percentile of LDL-C in both men and women. All statistical analyses were performed by SAS statistical software (V.9.4). All p values were two tailed, and the significance level was set at 0.05.

## RESULTS

### Baseline characteristics of the study population

A total of 4981 men and 5529 women were eligible for the final analysis. The median of LDL-C levels in women was significantly higher than that in men (p<0.0001, online supplementary figure S1). Compared with women, men were older, had smaller BMI values and possessed greater smoking rate and drinking rate (all p<0.0001). The prevalence rates of heart disease, arthritis and digestive disease in women were higher than those in men, but the prevalence rates of asthma and lung disease were lower in women (all p<0.0001, table 1).

### Characteristics of men and women according to the quintiles of LDL-C levels

After stratification by the quintiles of LDL-C levels, BMI, SBP, DBP and haemoglobin in men were elevated with ascending quintiles as a whole (all p<0.001) (table 2). The prevalence of HBS/diabetes was highest in the bottom quintile of LDL-C and lowest in the fourth quintile, with prevalence rates of 34.81% and 27.06%, respectively. There were no statistical differences among LDL-C quintiles for many other characteristics (eg, age, smoking, drinking, ADL disability, living alone, stroke, cancer, heart disease, lung disease, liver disease, kidney disease, digestive disease, asthma, arthritis, psychological problem and memory problem) (all p>0.05).

In women, LDL-C quintiles were positively associated with age, BMI, SBP, DBP and haemoglobin (all p<0.001) (table 3). The prevalence rates of HBS/diabetes and liver disease in women were significantly different among different LDL-C quintiles (all p<0.001). For the remaining variables (eg, smoking, drinking, household income, ADL disability, educational level, rural residence, hand grip strength, stroke, cancer, heart disease, lung disease, kidney disease, digestive disease, asthma, arthritis, psychological problem and memory problem), no differences were observed (all p>0.05).

### Associations of LDL-C levels with all-cause mortality

In men, 305 out of 4981 participants died during a 4-year follow-up. The mortality rates were declining with ascending quintiles (table 4). Compared with the first quintile, the univariate HRs (95%CIs) were 0.733 (0.533 to 1.007) for the second quintile; 0.639 (0.458 to 0.892) for the third quintile; 0.519 (0.364 to 0.739) for the fourth quintile and 0.512 (0.359 to 0.732) for the fifth quintile. After adjustment for a series of potential confounders, the non-linear association between LDL-C and all-cause mortality was observed. As compared with the first quintile, the multivariable HRs (95% CIs) were as follows: second quintile, 0.818 (0.531 to 1.260); third quintile, 0.782 (0.507 to 1.208); fourth quintile, 0.605 (0.381 to 0.962); fifth quintile, 0.803 (0.506 to 1.274). In women, there were 219 deaths during a 4-year follow-up. The mortality rate was highest in the first quintile. After adjustment for potential confounders, no quintile showed significant lower mortality rates compared with the first quintile (all p>0.05) (table 4).

The quintile analysis indicated that the relationship between LDL-C with all-cause mortality might be non-linear. Therefore, RCS was further performed to investigate the association between all-cause mortality and LDL-C on a continuous scale. The results from RCS showed that when the 20th percentile of LDL-C levels

**Table 2** Baseline characteristics of participants by quintiles of LDL-C in men

| Characteristics | Quintile 1 (n=991) (≤83.89 mg/dL) | Quintile 2 (n=1008) (83.89~101.68) | Quintile 3 (n=991) (101.68~117.14) | Quintile 4 (n=1004) (117.14~136.86) | Quintile 5 (n=987) (>136.86) | P value |
|---|---|---|---|---|---|---|
| Age-year | 59 (52~66) | 59 (53~67) | 59 (52~66) | 59 (53~67) | 59 (53~65) | 0.5405 |
| BMI-kg/m$^2$ | 21.57 (19.78~24.21) | 22.18 (20.31~24.40) | 22.36 (20.44~24.61) | 22.64 (20.65~25.07) | 23.27 (20.95~25.68) | <0.0001 |
| SBP-mm Hg | 126.00 (114.00~140.00) | 126.00 (115.33~139.67) | 127.00 (114.67~142.33) | 128.00 (115.67~142.50) | 130.67 (118.00~144.00) | 0.0002 |
| DBP-mm Hg | 75.00 (66.67~83.33) | 74.33 (66.67~82.00) | 74.67 (67.33~83.33) | 75.33 (67.67~83.67) | 76.67 (69.33~85.33) | 0.0002 |
| Hand grip strength (kg) | 35.93 (30.00~42.50) | 36.00 (30.00~42.60) | 36.93 (30.00~43.00) | 37.50 (31.50~43.75) | 37.50 (31.50~42.90) | 0.0132 |
| Above-average household income-no (%) | 402 (48.43) | 413 (47.91) | 407 (48.05) | 430 (49.20) | 452 (54.99) | 0.0186 |
| Educational level-no (%) | | | | | | 0.0035 |
| 1 | 870 (87.79) | 891 (88.39) | 821 (82.85) | 862 (85.86) | 843 (85.41) | |
| 2 | 105 (10.60) | 94 (9.33) | 145 (14.63) | 123 (12.25) | 125 (12.66) | |
| 3 | 16 (1.61) | 23 (2.28) | 25 (2.52) | 19 (1.93) | 19 (1.93) | |
| ADL disability-no (%) | 149 (15.27) | 163 (16.30) | 148 (15.10) | 141 (14.23) | 115 (11.81) | 0.0590 |
| Living alone-no (%) | 101 (10.19) | 112 (11.11) | 85 (8.58) | 94 (9.36) | 79 (8.00) | 0.1264 |
| Rural residence-no (%) | 659 (66.50) | 702 (69.64) | 634 (63.98) | 664 (66.14) | 623 (63.12) | 0.0216 |
| Lifestyle-no (%) | | | | | | |
| Smoking ever | 755 (76.57) | 754 (75.02) | 751 (75.94) | 759 (75.75) | 719 (72.92) | 0.3788 |
| Drinking ever | 654 (66.40) | 663 (66.10) | 664 (67.14) | 664 (66.40) | 652 (66.13) | 0.9890 |
| Disease history-no (%) | | | | | | |
| Hypertension | 359 (42.14) | 349 (39.48) | 385 (42.87) | 397 (44.51) | 414 (47.81) | 0.0092 |
| HBS/diabetes | 337 (34.81) | 277 (28.35) | 266 (27.85) | 266 (27.06) | 314 (32.78) | 0.0003 |
| Cancer | 13 (1.32) | 7 (0.70) | 5 (0.51) | 6 (0.60) | 9 (0.92) | 0.2728 |
| Stroke | 26 (2.64) | 33 (3.31) | 21 (2.13) | 25 (2.50) | 28 (2.85) | 0.5818 |
| Heart disease | 86 (8.78) | 98 (9.81) | 103 (10.47) | 113 (11.31) | 99 (10.10) | 0.4453 |
| Lung disease | 131 (13.37) | 135 (13.46) | 106 (10.77) | 120 (12.00) | 128 (13.09) | 0.3170 |
| Arthritis | 300 (30.43) | 332 (33.10) | 280 (28.34) | 312 (31.23) | 324 (33.03) | 0.1233 |
| Liver disease | 51 (5.23) | 42 (4.19) | 35 (3.57) | 38 (3.81) | 31 (3.18) | 0.1855 |
| Kidney disease | 56 (5.73) | 73 (7.29) | 59 (6.02) | 57 (5.71) | 64 (6.56) | 0.5518 |
| Digestive disease | 211 (21.46) | 213 (21.24) | 225 (22.82) | 203 (20.28) | 178 (18.14) | 0.1264 |
| Asthma | 66 (6.76) | 49 (4.89) | 43 (4.36) | 59 (5.91) | 66 (6.72) | 0.0754 |
| Psychological problem | 14 (1.43) | 10 (1.00) | 14 (1.42) | 14 (1.40) | 8 (0.82) | 0.6092 |
| Memory problem | 25 (2.55) | 13 (1.30) | 21 (2.13) | 19 (1.90) | 19 (1.94) | 0.3810 |
| Laboratory measurements | | | | | | |
| Triglyceride-mg/dL | 96.46 (65.49~177.88) | 88.50 (64.61~129.21) | 92.93 (69.03~138.95) | 97.35 (72.57~136.29) | 108.86 (79.65~152.22) | <0.0001 |
| HDL cholesterol-mg/dL | 47.55 (35.95~59.92) | 49.48 (39.43~60.70) | 47.93 (39.43~58.76) | 49.10 (39.82~59.54) | 48.71 (41.37~58.76) | 0.0061 |
| Haemoglobin-(g/L) | 148 (136~160) | 149 (138~161) | 150 (140~162) | 151 (140~163) | 154 (143~165) | <0.0001 |
| LDL-C-mg/dL | 71.91 (61.47~78.09) | 93.56 (88.92~97.81) | 109.41 (105.54~113.27) | 126.42 (121.39~131.44) | 153.87 (144.20~170.49) | <0.0001 |

ADL, activity of daily living; BMI, body mass index; DBP, diastolic blood pressure; HBS, high blood sugar; HDL, high-density lipoprotein; LDL-C, low-density lipoprotein cholesterol; SBP, systolic blood pressure.

**Table 3** Baseline characteristics of participants by quintiles of LDL-C in women

| Characteristics | Quintile 1 (n=1114) (≤91.24 mg/dL) | Quintile 2 (n=1102) (91.24~109.41) | Quintile 3 (n=1096) (109.41~126.03) | Quintile 4 (n=1111) (126.03~147.68) | Quintile 5 (n=1106) (>147.68) | P value |
|---|---|---|---|---|---|---|
| Age-year | 56 (49~64) | 56 (49~64) | 57 (50~63) | 58 (52~65) | 59 (53~66) | <0.0001 |
| BMI-kg/m$^2$ | 23.15 (20.90~25.76) | 23.30 (21.01~25.82) | 23.39 (21.10~26.02) | 23.63 (21.32~26.26) | 24.12 (21.57~26.83) | <0.0001 |
| SBP-mm Hg | 124.67 (112.00~141.00) | 125.67 (113.00~142.33) | 127.00 (114.67~143.00) | 127.67 (114.67~142.33) | 130.00 (117.00~146.00) | <0.0001 |
| DBP-mm Hg | 73.33 (65.67~81.67) | 74.33 (66.67~82.67) | 74.00 (67.33~82.67) | 74.67 (67.67~82.67) | 75.33 (67.67~83.33) | 0.0103 |
| Hand grip strength (kg) | 24.75 (19.80~30.00) | 25.00 (20.00~29.50) | 25.50 (20.55~30.00) | 24.95 (20.00~29.50) | 24.50 (20.25~29.50) | 0.3187 |
| Above-average household income-no (%) | 458 (48.21) | 482 (50.42) | 461 (48.99) | 497 (53.61) | 462 (50.44) | 0.1720 |
| Education level-no (%) | | | | | | 0.4079 |
| 1 | 1023 (91.83) | 1030 (93.47) | 1023 (93.34) | 1025 (92.26) | 1033 (93.40) | |
| 2 | 76 (6.82) | 67 (6.08) | 63 (5.75) | 76 (6.84) | 65 (5.88) | |
| 3 | 15 (1.35) | 5 (0.45) | 10 (0.91) | 10 (0.90) | 8 (0.72) | |
| ADL disability-no (%) | 216 (19.69) | 183 (16.90) | 188 (17.59) | 215 (19.58) | 209 (19.05) | 0.3416 |
| Living alone-no (%) | 155 (13.91) | 162 (14.70) | 143 (13.05) | 186 (16.74) | 186 (16.82) | 0.0429 |
| Rural residence-no (%) | 715 (64.18) | 695 (63.07) | 726 (66.24) | 706 (63.55) | 682 (61.66) | 0.2526 |
| Lifestyle-no (%) | | | | | | |
| Smoking ever | 71 (6.39) | 93 (8.49) | 78 (7.15) | 79 (7.14) | 99 (8.97) | 0.1294 |
| Drinking ever | 173 (15.61) | 160 (14.61) | 169 (15.50) | 165 (14.91) | 159 (14.40) | 0.9112 |
| Disease history-no (%) | | | | | | |
| Hypertension | 414 (41.69) | 440 (44.58) | 431 (44.21) | 451 (45.46) | 509 (50.70) | 0.0014 |
| HBS/diabetes | 310 (28.78) | 290 (27.46) | 280 (26.19) | 311 (28.88) | 373 (34.53) | 0.0003 |
| Cancer | 14 (1.27) | 15 (1.38) | 18 (1.66) | 14 (1.27) | 6 (0.54) | 0.1876 |
| Stroke | 31 (2.80) | 16 (1.47) | 27 (2.47) | 31 (2.81) | 31 (2.81) | 0.1906 |
| Heart disease | 144 (13.08) | 135 (12.41) | 140 (12.89) | 166 (15.09) | 153 (13.87) | 0.3873 |
| Lung disease | 110 (9.94) | 112 (10.29) | 103 (9.45) | 91 (8.27) | 92 (8.36) | 0.3566 |
| Arthritis | 454 (41.05) | 450 (41.21) | 421 (38.55) | 454 (41.05) | 464 (42.14) | 0.5189 |
| Liver disease | 53 (4.80) | 40 (3.69) | 25 (2.31) | 43 (3.91) | 27 (2.46) | 0.0060 |
| Kidney disease | 74 (6.71) | 62 (5.69) | 57 (5.24) | 66 (6.02) | 56 (5.09) | 0.4886 |
| Digestive disease | 279 (25.20) | 267 (24.45) | 269 (24.70) | 299 (27.13) | 279 (25.34) | 0.6323 |
| Asthma | 43 (3.89) | 43 (3.94) | 44 (4.03) | 39 (3.54) | 47 (4.26) | 0.9384 |
| Psychological problem | 16 (1.45) | 19 (1.74) | 23 (2.12) | 17 (1.54) | 19 (1.73) | 0.7943 |
| Memory problem | 18 (1.63) | 19 (1.74) | 11 (1.01) | 15 (1.36) | 23 (2.09) | 0.3245 |
| Laboratory measurements | | | | | | |
| Triglyceride-mg/dL | 110.63 (73.46~192.93) | 103.54 (74.34~153.99) | 107.97 (77.88~152.22) | 110.63 (82.31~151.34) | 125.23 (92.93~162.84) | <0.0001 |
| HDL cholesterol-mg/dL | 46.39 (35.57~57.60) | 50.64 (40.98~60.70) | 51.42 (42.53~60.70) | 51.80 (43.69~61.08) | 51.80 (44.46~60.31) | <0.0001 |
| Haemoglobin-(g/L) | 132 (121~143) | 133 (123~145) | 136 (126~147) | 137 (128~146) | 138 (128~148) | <0.0001 |
| LDL-C-mg/dL | 78.48 (68.04~85.83) | 100.90 (96.26~105.54) | 117.91 (113.66~121.78) | 135.70 (130.28~141.50) | 165.08 (155.03~179.00) | <0.0001 |

ADL, activity of daily living; BMI, body mass index; DBP, diastolic blood pressure; HBS, high blood sugar; HDL, high-density lipoprotein; LDL-C, low-density lipoprotein cholesterol; SBP, systolic blood pressure.

(84 mg/dL) was used as the reference, lower LDL-C (<84 mg/dL) was associated with higher risk of 4-year all-cause mortality in men, and moderately higher LDL-C (84–135 mg/dL) possessed lower total mortality risk, but the association was not statistically significant for much higher LDL-C concentrations (>135 mg/dL) (figure 2). The subgroup analyses by age indicated that when the 20th percentile of LDL-C levels was taken as the reference, a lower level of LDL-C was associated with a higher risk of 4 years all-cause mortality in both middle-aged (45–60 years) and elderly (≥60 years) men (figure 3). For women, LDL-C was not significantly associated with 4 years all-cause mortality (figures 2 and 3).

In addition, we found that 125 out of 4981 men and 89 out of 5529 women had LDL-C <50 mg/dL. When participants with LDL-C <50 mg/dL were excluded,

**Table 4** Associations between all-cause mortality and LDL-C

|  | Total | Deaths (%) | Unadjusted | | Adjusted* | |
|---|---|---|---|---|---|---|
|  |  |  | HR (95% CI) | P value | HR (95% CI) | P value |
| **Men** | | | | | | |
| Q1 | 991 | 88 (8.88) | 1 | – | 1 | – |
| Q2 | 1008 | 67 (6.65) | 0.733 (0.533~1.007) | 0.0554 | 0.818 (0.531~1.260) | 0.3619 |
| Q3 | 991 | 57 (5.75) | 0.639 (0.458~0.892) | 0.0084 | 0.782 (0.507~1.208) | 0.2677 |
| Q4 | 1004 | 47 (4.68) | 0.519 (0.364~0.739) | 0.0003 | 0.605 (0.381~0.962) | 0.0335 |
| Q5 | 987 | 46 (4.66) | 0.512 (0.359~0.732) | 0.0002 | 0.803 (0.506~1.274) | 0.3520 |
| **Women** | | | | | | |
| Q1 | 1114 | 52 (4.67) | 1 | – | 1 | – |
| Q2 | 1102 | 49 (4.45) | 0.960 (0.650~1.419) | 0.8394 | 1.245 (0.749~2.071) | 0.3985 |
| Q3 | 1096 | 29 (2.65) | 0.565 (0.359~0.890) | 0.0138 | 0.626 (0.345~1.136) | 0.1233 |
| Q4 | 1111 | 41 (3.69) | 0.792 (0.526~1.192) | 0.2632 | 0.852 (0.489~1.483) | 0.5704 |
| Q5 | 1106 | 48 (4.34) | 0.926 (0.625~1.371) | 0.7007 | 0.958 (0.563~1.630) | 0.8736 |

*Adjusted for age, smoking, drinking, BMI, living alone status, household income, educational level, rural residence, ADL disability, hand grip strength, HDL-C, triglyceride, haemoglobin, hypertension, HBS/diabetes, history of stroke, cancer, heart disease, lung disease, liver disease, kidney disease, digestive disease, asthma, arthritis, psychological problem and memory problem.
ADL, activity of daily living; BMI, body mass index; HBS, high blood sugar; HDL-C, high-density lipoprotein cholesterol.

the HR of the fourth LDL-C quintile in men was changed a little with marginal statistical significance (p=0.0698, online supplementary table S1). Moreover, no interactions were found between LDL-C and potential risk factors of mortality, with the exception that the interaction between LDL-C and smoking in

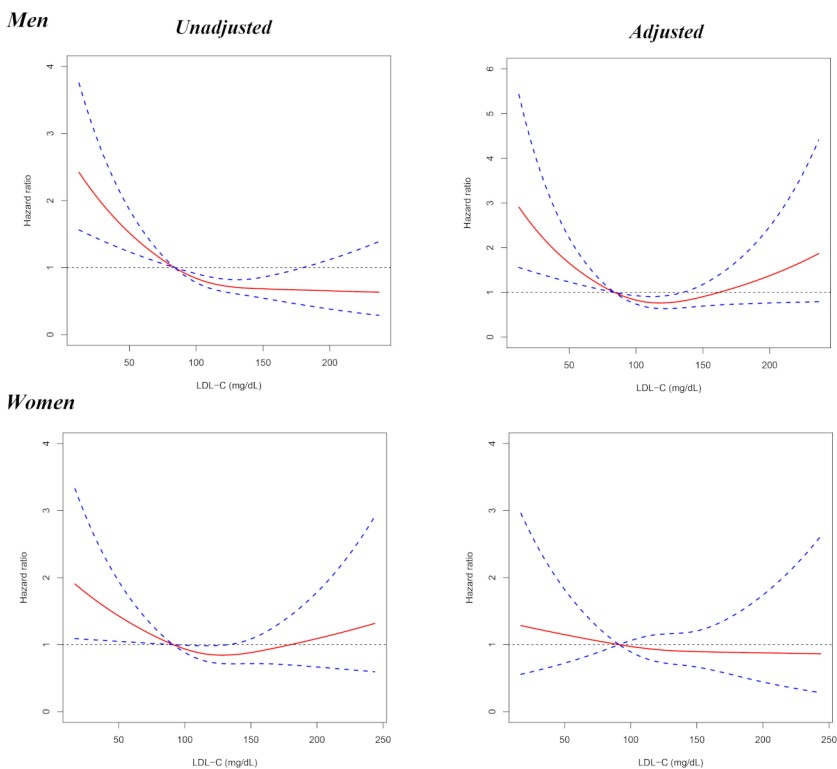

**Figure 2** Results from restricted cubic splines for the association between LDL-C and 4-year all-cause mortality in men and women, respectively. The multivariable models were adjusted for age, smoking, drinking, BMI, living alone status, household income, educational level, rural residence, ADL disability, hand grip strength, HDL-C, triglyceride, haemoglobin, hypertension, HBS/diabetes, history of stroke, cancer, heart disease, lung disease, liver disease, kidney disease, digestive disease, asthma, arthritis, psychological problem and memory problem. ADL, activity of daily living; BMI, body mass index; HBS, high blood sugar; HDL-C, high-density lipoprotein cholesterol; LDL-C, low-density lipoprotein cholesterol.

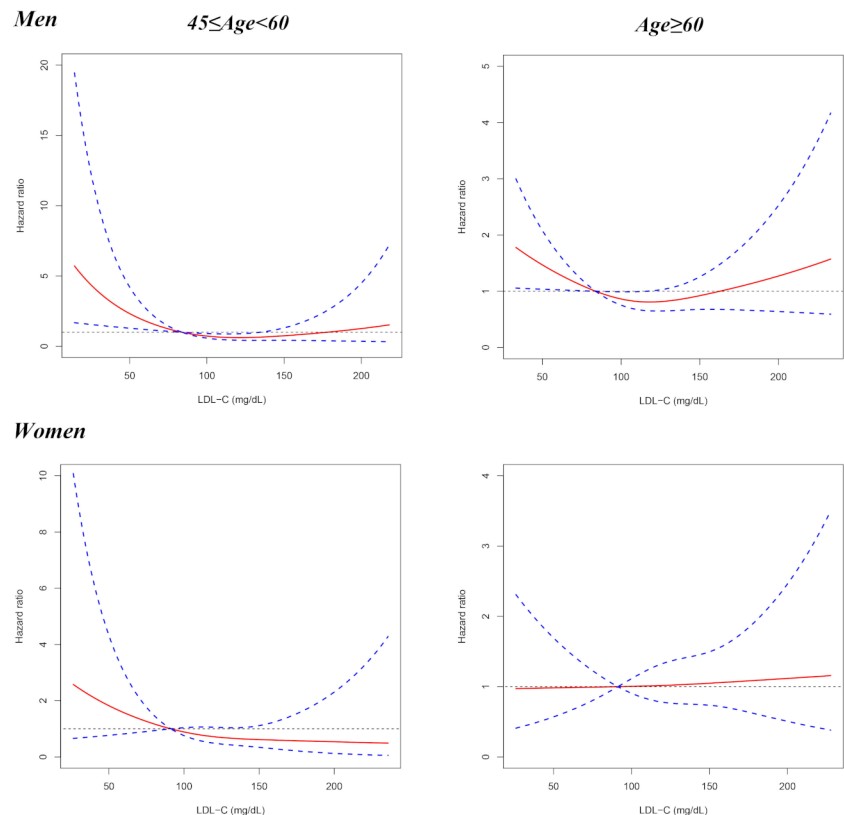

**Figure 3** Results from restricted cubic spline for the association between LDL-C and 4 years all-cause mortality for middle-aged (45–60 years old) and elderly (≥60 years old) people, respectively. The multivariable models were adjusted for age, smoking, drinking, BMI, living alone status, household income, educational level, rural residence, ADL disability, hand grip strength, HDL-C, triglyceride, haemoglobin, hypertension, HBS/diabetes, history of stroke, cancer, heart disease, lung disease, liver disease, kidney disease, digestive disease, asthma, arthritis, psychological problem and memory problem. ADL, activity of daily living; BMI, body mass index; HBS, high blood sugar; HDL-C, high-density lipoprotein cholesterol; LDL-C, low-density lipoprotein cholesterol.

women was statistical significant (p=0.0498, online supplementary table S2).

## DISCUSSION

In this study, we investigated the relationship between LDL-C and 4 years all-cause mortality among the middle-aged and elderly Chinese population. In men, a very low level of LDL-C was associated with increased mortality risk. In women, LDL-C was not significantly associated with 4 years all-cause mortality.

LDL has been well established as an important cause of cardiovascular disease (CVD) for decades.[1] Since CVD is the leading cause of mortality throughout the world, it is logically reasonable that increased LDL-C should contribute to increased CVD mortality and possibly all-cause mortality. Indeed, evidences from prospective epidemiological studies showed a positive association between non-HDL-C concentration and ischaemic heart disease mortality.[8] However, non-HDL-C includes both LDL-C and very LDL-C. It is surprising that the direct association of LDL-C with CVD mortality was not consistently reported among studies. Abdullah *et al*. demonstrated that LDL-C was independently associated with CVD mortality in a low 10-year risk cohort with long-term follow-up.[9] By

contrast, Tikhonof *et al*. reported that the elderly subjects (≥65 years) possessed the highest CVD mortality in the lowest LDL-C quartile.[10] Meanwhile, many other studies also found no association between LDL-C and CVD mortality.[11–13] When the results about the association of LDL-C with CVD mortality were inconsistent, it is more surprising to find that few studies have reported the positive association between LDL-C and all-cause mortality. In the study by Abdullah *et al*, there were already no associations or minimal positive associations between high LDL-C categories and all-cause mortality even in univariable Cox analyses.[9] Most remarkably, results of multivariable Cox analyses in this study were not provided for all-cause mortality, which could exert a substantial impact on the final association.[14] Actually, a large number of studies reported no association or even an inverse association between LDL-C and all-cause mortality, which has been summarised in a systematic review by Ravnskov *et al*.[5] Therefore, very low LDL-C in populations not on lipid therapy may be associated with harm.

It should be noted that the confounding effect of statin treatment should be minimised, as this study excluded those who used lipid-lowering interventions. There was also no association between baseline LDL-C and the

presence of cancer, stroke and heart disease in men and women (tables 2 and 3). Moreover, when participants who had died during the first observation year were excluded, this relationship was not changed (online supplementary table S3 and figure S2). This could relieve the concern that serious diseases may lower cholesterol soon before death occurs. One of the possible reasons for the difference between men and women may be due to fewer death events in women than in men, which might result in insufficient power for the association.

Several explanations for the unfavourable effects of low LDL-C levels may be proposed. LDL-C has been suggested to play an important role in host defence against both bacterial and viral pathogens.[15] Indeed, many animal and laboratory experiments have shown that LDL could bind to and inactivate a broad range of microorganisms and their toxic products.[16–18] This hypothesis may be further supported by the recent finding that LDL-C was associated with reduced infectious mortality based on the data from 37 250 patients in the international Monitoring Dialysis Outcomes database.[2] In addition, it has been proposed that LDL-C may have the potential to protect against cancer as many cancer types are caused by viruses.[19] Ravnskov et al[20] reviewed nine cohort studies including more than 140 000 individuals followed for 10–30 years and found that low cholesterol was associated with cancer.[20] Moreover, cholesterol-lowering experiments on rodents have led to cancer as well.[21] In agreement with these findings, individuals with familial hypercholesterolaemia have been found to possess significantly lower cancer mortality.[22] Therefore, lower LDL-C may contribute to a higher risk of death from infection and cancer, which in turn results in increased all-cause mortality.

This study demonstrated that middle-aged and elderly Chinese men with very low LDL-C had an increased risk of all-cause mortality, which calls for special attention to be paid to the possible harmful effect of a very low level of LDL-C. However, some limitations should be noted. First, the follow-up period was limited to 4 years. For a longer follow-up time, the associations between LDL-C and all-cause mortality in women might be displayed. Second, cause-specific mortality data were not available for the time being, preventing the analysis of the association between LDL-C and cause-specific mortality. Third, there are issues of multiple testing for comparisons of characteristics among LDL-C quintiles, which could result in type I error inflation. Fourth, some of the measured comorbidities were not specified and detailed in the database, such as lung disease, digestive disease, liver disease, kidney disease, psychological problem and memory problem. At last, well-designed, large-scale population studies are needed to formulate the specific LDL-C level(s) threshold for mortality risk in the future.

In China, compared with moderately elevated LDL-C (eg, 117–137 mg/dL), a lower plasma level of LDL-C (eg, ≤84 mg/dL) was associated with an increased risk of 4 years all-cause mortality in middle-aged and elderly men. The findings in this study may suggest the potential harmful effect of a quite low level of LDL-C. More prospective and well-designed studies are needed to validate the relationship between LDL-C and mortality.

**Author affiliations**
[1]Liyang Center for Disease Control and Prevention, Liyang, Jiangsu, China
[2]State Key Laboratory of Organ Failure Research, Department of Biostatistics, Guangdong Provincial Key Laboratory of Tropical Disease Research, School of Public Health, Southern Medical University, Guangzhou, Guangdong, China
[3]Medical College of Soochow University, Suzhou, Jiangsu, China
[4]Department of Epidemiology and Biostatistics, School of Public Health, Medical College of Soochow University, Suzhou, Jiangsu, China

**Acknowledgements** This analysis uses data or information from the Harmonised CHARLS dataset and Codebook, Version C as of April 2018 developed by the Gateway to Global Ageing Data. The development of the Harmonised CHARLS was funded by the National Institute on Ageing (R01 AG030153, RC2 AG036619, R03 AG043052). For more information, please refer to www.g2aging.org.

**Contributors** CK and YS conceived and designed the research; LZ and CK wrote the manuscript; and YW and SY performed the data analysis. All authors contributed to the interpretations of the findings. All authors reviewed the manuscript.

**Funding** This work was supported by National Natural Science Foundation of China (81 703 316 to CK, 81 703 322 to YW) and Natural Science Foundation of Jiangsu Province (BK20170350).

**Competing interests** None declared.

**Patient and public involvement** Patients and/or the public were not involved in the design, or conduct, or reporting, or dissemination plans of this research.

**Patient consent for publication** Not required.

**Ethics approval** This study was approved by Biomedical Ethics Review Committee of Peking University.

**Provenance and peer review** Not commissioned; externally peer reviewed.

**Data availability statement** Data are available in a public, open access repository. This study used data from the China Health and Retirement Longitudinal Study (http://charls.pku.edu.cn/)

**ORCID iD**
Chaofu Ke http://orcid.org/0000-0003-3188-4307

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
