## [Reviewer comments · BMJ Open]

ARTICLE DETAILS

TITLE (PROVISIONAL)	Low-density lipoprotein cholesterol and all-cause mortality: findings from the China Health and Retirement Longitudinal Study
AUTHORS	Zhou, Liang; Wu, Ying; Yu, Shaobo; shen, yueping; Ke, Chaofu

VERSION 1 – REVIEW

REVIEWER	Shuaib Abdullah University of Texas Southwestern Medical Center USA
REVIEW RETURNED	11-Feb-2020

GENERAL COMMENTS	Zhou et al. evaluate the association of LDL-C with all-cause mortality in 4983 men and 5535 women of the CHARLS study aged > 45 years and not on lipid-lowering therapy with median follow-up of 4 years. In men, compared to those with LDL-C in the 1st quintile, quintiles 4 and 5 had significantly lower all-cause mortality in multivariable Cox regression analyses. No significant association was seen in women. The manuscript is of interest in that while most experts have accepted the 'LDL hypothesis' and LDL-C and non-HDL-C have been demonstrated to be associated with CVD outcomes for decades, several recent studies have demonstrated a lack of association, and even inverse association between LDL-C and all cause mortality, especially in elderly populations. The manuscript is well written. However, there are a number of limitations to the current analyses, including the short follow-up period and the lack of confirmation of cause specific mortality, which the authors acknowledge. There are a number of additional issues with the manuscript that need to be addressed. 1) Most of the previous studies that have demonstrated the lack of beneficial association of LDL-C and all-cause mortality examined elderly populations, and previous studies with middle-aged subjects have mostly shown a direct association of LDL-C with CVD outcomes. The current study consists of middle-aged and elderly patients. The authors should assess whether age modifies the association between LDL-C and all-cause mortality by performing stratified analyses in those </> 60 years old and by checking for interactions. 2) It is surprising that so many patients with diabetes mellitus (30%) and established cardiovascular disease (10-13%) were not on statins, which have been shown to be beneficial and are recommended in these patient populations by the current guidelines. Statistical interactions should be assessed between LDL-C and other risk factors including diabetes mellitus, CVD, lung disease tobacco use, alcohol use, obesity, etc.
--

	3) Based on their current results, the authors hypothesize that LDL-C lowering may have unfavorable effects on all-cause mortality. How do the authors reconcile their results with the fact that multiple statin and PCSK9 inhibitor trials with similar follow-up periods and patient ages have demonstrated either no significant effects on all-cause mortality, or in some cases beneficial effects? 4) In addition to cause-specific mortality not being able to be assessed in the current cohort, it is not clear what the significance of many of the measured co-morbidities is. For example, it is unclear what is meant by lung disease or how many of those that use alcohol are heavy drinkers. Is it possible to improve the granularity of the comorbidities? If not, this should also be listed as a limitation. 5) One of the explanations for why very low LDL-C may be related to all-cause mortality is that it may be confounded by significant terminal comorbidities. Since cause-specific mortality is not assessed, additional variables measured in the cohort such as socioeconomic status, hand grip strength, quality of life measurements (EuroQol), social interaction, albumin, and hemoglobin may be adjusted in multivariable analyses to account for evidence of terminal illness or frailty. 6) Similarly, it appears from the restricted cubic spline analyses that the highest risk of all-cause mortality is related to very low LDL-C, well below the 83.9 mg/dL cut-off for the first quintile. The median and interquartile range of each LDL-C quintile should be listed. How many of the patients in the lowest quintile had a LDL-C < 50 mg/dL? Does excluding these patients affect the odds ratios of the higher LDL-C quintiles? 7) The supplemental tables should be moved to the main part of the manuscript. 8) The inclusion criteria of the CHARLS study includes age > 45 years old. Why were there 458 patients with age < 45 in this study? 9) HDL-C and triglycerides should be accounted in the multivariable models. 10) Similar analyses should be done with non-HDL-C cholesterol to see if higher quintiles are associated with all-cause mortality. 11) The odds ratio and 95% confidence intervals in the abstract and results are slightly different than what is listed in the table. 12) Were subjects asked about starting lipid medications during follow-up visits? 13) In the Discussion section, it is stated that the reason this association was not seen in women was because women had much lower LDL-C, when in fact LDL-C was only about 8 mg/dL lower per quintile cut-offs, which is not clinically significant. In addition, it is stated that there is a high risk trend for women at low LDL-C. What data is this statement being based upon? 14) Further data on the parameters for the restricted cubic spline analyses is needed in the methods section, with a statistical review. 15) In the Discussion section, it is stated that Landmark analyses was performed excluding those patients that died during the first year. However, these results should be mentioned in the Results section and data shown in Supplemental Tables and Figures. 16) In the first paragraph of the Discussion section and in the abstract, it is stated that low LDL-C is associated with increased all-cause mortality risk. However, analyses are based on using low LDL-C as a reference and demonstrating a lower all-cause mortality risk in the 2 highest quintiles. It should be clarified what LDL-C values are and compared to what reference the risk differs.
--	---

REVIEWER	Robert M West University of Leeds
-----------------	--------------------------------------

	UK
REVIEW RETURNED	12-Mar-2020

GENERAL COMMENTS	This is an interesting manuscript and I encourage the authors to revise and resubmit. I have some concerns regarding the statistical methods. The analysis includes only those participants in the cohort who have LDL measurements at baseline, so that 34% are excluded. There should be some assessment of any potential bias that this might promote. Note that the findings are contentious so that extra care should be taken to investigate any potential bias. The wording regarding follow up is not clear to me. I assume though that there are some participants with LDL at baseline that are lost to follow up and then excluded. Again there is potential for bias and I would expect some effort to include at least a sensitivity analysis to explore the potential bias from this. A Cox model has been used. There appears to no checking of the proportional hazards assumption. Please provide this. It may reveal that proportional hazards are not valid. It is good to see that nonlinear effects of LDL are examined through use of restricted cubic splines. There are likely however to be nonlinear effects of age and there could be statistical interaction between many of the covariates. These have not been investigated. I would like more detail of what is meant by HBS/DM. The rates cited are high and I assume relate to diabetes and pre-diabetes taken together. This would then be consistent with other studies in China. Interpretation is based on $p < 0.05$ from, for example Table S2. There are issues of multiple testing and some qualification of this approach might be given in the study limitations. The authors acknowledge that there are issues around using all-cause mortality as the outcome. They are open to the possibility that LDL might be associated in opposite directions for different causes of mortality. This is true for other covariates also, and is a frustration of interpretation whenever a composite outcome is used.
---

VERSION 1 – AUTHOR RESPONSE

Reviewer 1

1. Most of the previous studies that have demonstrated the lack of beneficial association of LDL-C and all-cause mortality examined elderly populations, and previous studies with middle-aged subjects have mostly shown a direct association of LDL-C with CVD outcomes. The current study consists of middle-aged and elderly patients. The authors should assess whether age modifies the association between LDL-C and all-cause mortality by performing stratified analyses in those 60 years old and by checking for interactions.

Response: Thank you very much for your suggestion. We have performed stratified analyses in those Figure 1 Results from restricted cubic splines for the association between LDL-C and 4-year all-cause mortality for middle-aged (45~60 years old) and elderly (≥ 60 years old) people, respectively. The multivariable models were adjusted for age, smoking, drinking, BMI, marital status, household

income, educational level, rural residence, ADL disability, HDL-C, triglyceride, hemoglobin, hypertension, HBS/diabetes, history of stroke, cancer, heart disease, lung disease, liver disease, kidney disease, digestive disease, asthma, arthritis, psychological problem and memory problem.

2. It is surprising that so many patients with diabetes mellitus (30%) and established cardiovascular disease (10-13%) were not on statins, which have been shown to be beneficial and are recommended in these patient populations by the current guidelines.

Response: Thank you for this comment. We have checked the data, and found that those who reported dyslipidemia accounted for 6.08%, 5.5% and 6.6% in the whole, male and female populations, respectively. The low awareness rates of dyslipidemia might explain the low proportion of lipid-lowering interventions in this cohort. In addition, as high blood sugar (HBS)/diabetes covers both diabetes and pre-diabetes, the prevalence rates in men (30.16%) and women (29.19%) are relatively high. Since the interview question ("Have you been diagnosed with HBS/DM by a doctor?") did not distinguish HBS from diabetes, we could not define diabetes alone in this study.

3. Statistical interactions should be assessed between LDL-C and other risk factors including diabetes mellitus, CVD, lung disease tobacco use, alcohol use, obesity, etc.

Response: According to your suggestion, we have checked the interactions between LDL-C and other risk factors. The results showed that no interactions were found between LDL-C and other risk factors, with the exception that the interaction between LDL-C and smoking in women was marginally significant (Table 1).

Table 1 Analyses of interactions between LDL-C and risk factors

ID Interaction term P value in men P value in women

1	LDL-C*obesity	0.2972	0.3615
2	LDL-C*rural residence	0.1110	0.8224
3	LDL-C*ADL disability	0.4685	0.3462
4	LDL-C*smoking	0.5806	0.0394
5	LDL-C*drinking	0.9719	0.4802
6	LDL-C*hypertension	0.7614	0.9823
7	LDL-C*diabetes	0.6596	0.1295
8	LDL-C*heart disease	0.7774	0.8128
9	LDL-C*stroke	0.1523	0.2814
10	LDL-C*cancer	0.7232	0.8481
11	LDL-C*lung disease	0.6097	0.5762
12	LDL-C*memory disease	0.6670	0.1806
13	LDL-C*kidney disease	0.1601	0.1078
14	LDL-C*arthritis	0.3237	0.1869
15	LDL-C*asthma	0.3453	0.5406
16	LDL-C*liver disease	0.6433	0.7163
17	LDL-C*digestive disease	0.0965	0.2415
18	LDL-C*psychological disease	0.4119	0.3283

4. Based on their current results, the authors hypothesize that LDL-C lowering may have unfavorable effects on all-cause mortality. How do the authors reconcile their results with the fact that multiple statin and PCSK9 inhibitor trials with similar follow-up periods and patient ages have demonstrated either no significant effects on all-cause mortality, or in some cases beneficial effects?

Response: Thank you very much for your comment. Indeed, there were many randomized clinical trials (RCTs) supporting the use of statin or PCSK9 inhibitor to lower the LDL-C levels. It should be mentioned that most of the study populations in these RCTs were high-risk populations of cardiovascular diseases but not the general population¹⁻³. Moreover, it must be admitted that our results only suggested the unfavorable effects of very low LDL-C levels on 4-year total mortality in middle-aged and elderly men. Based on the results from our study, it is still hard to determine whether

participants with very high LDL-C are associated with all-cause mortality risk in the middle-aged and elderly Chinese population.

5. In addition to cause-specific mortality not being able to be assessed in the current cohort, it is not clear what the significance of many of the measured co-morbidities is. For example, it is unclear what is meant by lung disease or how many of those that use alcohol are heavy drinkers. Is it possible to improve the granularity of the comorbidities? If not, this should also be listed as a limitation.

Response: Thank you for your suggestion. Some of the measured co-morbidities were not specified and detailed in the database, such as lung disease, digestive disease, liver disease, kidney disease, psychological problem and memory problem. We agree with your point that this should be listed as a limitation. In the revised manuscript, this limitation has been added.

6. One of the explanations for why very low LDL-C may be related to all-cause mortality is that it may be confounded by significant terminal comorbidities. Since cause-specific mortality is not accessed, additional variables measured in the cohort such as socioeconomic status, hand grip strength, quality of life measurements (EuroQol), social interaction, albumin, and hemoglobin may be adjusted in multivariable analyses to account for evidence of terminal illness or frailty.

Response: Based on your suggestion, data availability and biological plausibility, we have adjusted age, smoking, drinking, BMI, living alone, household income, educational level, rural residence, activity of daily living (ADL) disability, HDL-C, triglyceride, hemoglobin, hypertension, HBS/diabetes, history of stroke, cancer, heart disease, lung disease, liver disease, kidney disease, digestive disease, asthma, arthritis, psychological problem and memory problem in multivariable analyses. Except for the phenomenon that the highest quintile in men was not statistically significant, the other results were not changed a lot (Table 2 and Figure 2), and the association between very low LDL-C and high all-cause mortality in men persisted.

Table 2 Associations between all-cause mortality and LDL-C

Total Deaths (%) Unadjusted Adjusted*

HR (95%CI) P value HR (95%CI) P value

Men

Q1 991 88(8.88) 1 - 1 -

Q2 1008 67(6.65) 0.733(0.533~1.007) 0.0554 0.866(0.567~1.325) 0.5079

Q3 991 57(5.75) 0.639(0.458~0.892) 0.0084 0.782(0.507~1.206) 0.2651

Q4 1004 47(4.68) 0.519(0.364~0.739) 0.0003 0.577(0.363~0.916) 0.0197

Q5 987 46(4.66) 0.512(0.359~0.732) 0.0002 0.788(0.497~1.248) 0.3093

Women

Q1 1114 52(4.66) 1 - 1 -

Q2 1102 49(4.45) 0.960(0.650~1.419) 0.8394 1.348(0.816~2.229) 0.2440

Q3 1096 29(2.64) 0.565(0.359~0.890) 0.0138 0.675(0.375~1.214) 0.1889

Q4 1111 41(3.69) 0.792(0.526~1.192) 0.2632 0.974(0.567~1.674) 0.9239

Q5 1106 48(4.34) 0.926(0.625~1.371) 0.7007 1.043(0.620~1.755) 0.8736

*Adjusted for age, smoking, drinking, BMI, living alone, household income, educational level, rural residence, ADL disability, HDL-C, triglyceride, hemoglobin, hypertension, HBS/diabetes, history of stroke, cancer, heart disease, lung disease, liver disease, kidney disease, digestive disease, asthma, arthritis, psychological problem and memory problem.

Figure 2 Results from restricted cubic splines for the association between LDL-C and 4-year all-cause mortality in men and women, respectively. The multivariable models were adjusted for age, smoking, drinking, BMI, marital status, household income, educational level, rural residence, ADL disability, HDL-C, triglyceride, hemoglobin, hypertension, HBS/diabetes, history of stroke, cancer, heart disease, lung disease, liver disease, kidney disease, digestive disease, asthma, arthritis, psychological problem and memory problem.

7. Similarly, it appears from the restricted cubic spline analyses that the highest risk of all-cause mortality is related to very low LDL-C, well below the 83.9 mg/dL cut-off for the first quintile. The median and interquartile range of each LDL-C quintile should be listed.

Response: We have listed the median and interquartile range of each LDL-C quintile in the revised manuscript (Table 3).

Table 3 Median and interquartile range of each LDL-C quintile

LDL-C quintile	Men	Women
Quintile 1	71.91 (61.47~78.09)	78.48 (68.04~85.83)
Quintile 2	93.56 (88.92~97.81)	100.90 (96.26~105.54)
Quintile 3	109.41 (105.54~113.27)	117.91 (113.66~121.78)
Quintile 4	126.42 (121.39~131.44)	135.70 (130.28~141.50)
Quintile 5	153.87 (144.20~170.49)	165.08 (155.03~179.00)

8. How many of the patients in the lowest quintile had a LDL-C < 50 mg/dL? Does excluding these patients affect the odds ratios of the higher LDL-C quintiles?

Response: 125 out of 4981 men and 89 out of 5529 women had a LDL-C < 50mg/dL. When those with LDL-C < 50 mg/dL were excluded, the odds ratio of the fourth LDL-C quintile in men was a little changed but still with statistical significance (Table 4).

Table 4 Associations between all-cause mortality and LDL-C#

Total Deaths (%)	Unadjusted		Adjusted*	
	HR (95%CI)	P value	HR (95%CI)	P value
Men				
Q1 866 75 (8.66)	1	-	-	-
Q2 1008 67(6.65)	0.750(0.539~1.042)	0.0865	0.915(0.588~1.423)	0.6925
Q3 991 57(5.75)	0.653(0.463~0.922)	0.0155	0.831(0.530~1.303)	0.4191
Q4 1004 47(4.68)	0.531(0.368~0.764)	0.0007	0.611(0.379~0.985)	0.0432
Q5 987 46(4.66)	0.524(0.363~0.756)	0.0006	0.843(0.522~1.359)	0.4827
Women				
Q1 1025 46 (4.49)	1	-	-	-
Q2 1102 49(4.45)	0.998(0.667~1.492)	0.9920	1.318(0.791~2.195)	0.2887
Q3 1096 29(2.64)	0.587(0.369~0.935)	0.0248	0.662(0.366~1.196)	0.1718
Q4 1111 41(3.69)	0.822(0.540~1.253)	0.3628	0.952(0.551~1.645)	0.8612
Q5 1106 48(4.34)	0.962(0.642~1.441)	0.8511	1.012(0.597~1.716)	0.9637

#Participants with LDL-C < 50 mg/dL were excluded.

*Adjusted for age, smoking, drinking, BMI, living alone, household income, educational level, rural residence, ADL disability, HDL-C, triglyceride, hemoglobin, hypertension, HBS/diabetes, history of stroke, cancer, heart disease, lung disease, liver disease, kidney disease, digestive disease, asthma, arthritis, psychological problem and memory problem.

9. The supplemental tables should be moved to the main part of the manuscript.

Response: We have moved the supplementary tables to the main part of the manuscript.

10. The inclusion criteria of the CHARLS study includes age > 45 years old. Why were there 458 patients with age < 45 in this study?

Response: We have rechecked the dataset from CHARLS, and found that 458 patients were with age < 45 and 26 with missing values in this study. The baseline national wave of CHARLS is being fielded in 2011 and includes about 10,257 households and 17,708 individuals in 150 counties/districts and 450 villages/resident committees⁴. For each household, there are at least one family member aged 45 years old and over. If there more than 1 family member aged 45 years and over, one family member aged 45 years and over is randomly selected as the main respondent. Both the main respondents and their spouses (regardless of their age) form the 17,708 individuals. Therefore, the specific cohort design and sampling strategy result in some subjects (spouses) aged <45 years old.

To ensure a clear definition for the middle-aged and elderly population, we excluded those aged < 45 years or with missing values (n=484) in the revised manuscript.

11. HDL-C and triglycerides should be accounted in the multivariable models.

Response: Thank you for your suggestion. HDL-C and triglycerides have been adjusted in multivariable analyses.

12. Similar analyses should be done with non-HDL-C cholesterol to see if higher quintiles are associated with all-cause mortality.

Response: Thank you for your suggestion. We performed additional analyses on non-HDL-C. In both men and women, a very low level of non-HDL-C was associated with high mortality risk (Table 4 and Figure 3).

Table 4 Associations between all-cause mortality and non-HDL-C

Total Deaths (%)		Unadjusted		Adjusted*	
HR (95%CI)	P value	HR (95%CI)	P value	HR (95%CI)	P value
Men					
Q1	989 90 (9.10)	1	-	-	-
Q2	984 68 (6.91)	0.750(0.547~1.027)	0.0728	0.855(0.565~1.292)	0.4571
Q3	980 49 (5.00)	0.547(0.386~0.774)	0.0007	0.606(0.385~0.952)	0.0297
Q4	995 45 (4.52)	0.491(0.343~0.702)	<0.0001	0.648(0.406~1.033)	0.0684
Q5	985 52 (5.58)	0.573(0.407~0.806)	0.0014	0.840(0.519~1.357)	0.4755
Women					
Q1	1094 55 (5.03)	1	-	-	-
Q2	1095 44 (4.02)	0.792(0.533~1.178)	0.2502	0.787(0.475~1.305)	0.3534
Q3	1097 27 (2.46)	0.484(0.305~0.767)	0.0020	0.482(0.271~0.858)	0.0131
Q4	1097 37 (3.37)	0.663(0.437~1.006)	0.0536	0.648(0.375~1.121)	0.1209
Q5	1095 55 (5.02)	0.993(0.683~1.443)	0.9698	0.919(0.547~1.546)	0.7511

*Adjusted for age, smoking, drinking, BMI, living alone, household income, educational level, rural residence, ADL disability, HDL-C, triglyceride, hemoglobin, hypertension, HBS/diabetes, history of stroke, cancer, heart disease, lung disease, liver disease, kidney disease, digestive disease, asthma, arthritis, psychological problem and memory problem.

Figure 3 Results from restricted cubic splines for the association between non-HDL-C and 4-year all-cause mortality in men and women, respectively. The multivariable models were adjusted for age, smoking, drinking, BMI, living alone, household income, educational level, rural residence, ADL disability, HDL-C, triglyceride, hemoglobin, hypertension, HBS/diabetes, history of stroke, cancer, heart disease, lung disease, liver disease, kidney disease, digestive disease, asthma, arthritis, psychological problem and memory problem.

13. The odds ratio and 95% confidence intervals in the abstract and results are slightly different than what is listed in the table.

Response: In the revised manuscript, we have ensured the consistency of these findings in the abstract and results.

14. Were subjects asked about starting lipid medications during follow-up visits?

Response: We have rechecked the database, and found that lipid-lowering interventions were asked during follow-up visits. These interventions included taking Chinese traditional medicine, taking western modern medicine and other treatments. During follow-up, 986 individuals of the study population in our study began lipid-lowering interventions. After further excluding those 986 individuals, the findings and conclusion were not changed (Figure 4).

Figure 4 Results from restricted cubic splines for the association between LDL-C and 4-year all-cause

mortality in men and women (excluding those with lipid-lowering interventions during follow-up). The multivariable models were adjusted for age, smoking, drinking, BMI, living alone, household income, educational level, rural residence, ADL disability, HDL-C, triglyceride, hemoglobin, hypertension, HBS/diabetes, history of stroke, cancer, heart disease, lung disease, liver disease, kidney disease, digestive disease, asthma, arthritis, psychological problem and memory problem.

15. In the Discussion section, it is stated that the reason this association was not seen in women was because women had much lower LDL-C, when in fact LDL-C was only about 8 mg/dL lower per quintile cut-offs, which is not clinically significant. In addition, it is stated that there is a high risk trend for women at low LDL-C. What data is this statement being based upon?

Response: Thank you for your comment, and we agree with your point that this speculation was not that much reasonable. In the revised manuscript, we revised this discussion as follows: One of the possible reasons for the difference between men and women may be due to fewer death events in women than in men, which might result in insufficient statistical power for the association.

16. Further data on the parameters for the restricted cubic spline analyses is needed in the methods section, with a statistical review.

Response: Thank you for your suggestion. The association between all-cause mortality and LDL-C on a continuous scale was examined using restricted cubic splines (RCS) incorporated in Cox proportional hazards models. Bayesian Information Criterion (BIC) was used to determine the optimal number of knots in RCS. In this study, 3 knots were used in all RCS analyses, with knot locations at the 10th, 50th, and 90th percentiles of LDL-C. To be consistent with quintile analyses, the reference point was the 20th percentile of LDL-C in both men and women. In the revised manuscript, we have provided the details for RCS analyses.

17. In the Discussion section, it is stated that Landmark analyses was performed excluding those patients that died during the first year. However, these results should be mentioned in the Results section and data shown in Supplemental Tables and Figures.

Response: According to your suggestion, we have included these results as Supplementary table and figure (Table 5 and Figure 5).

Table 5 Associations between all-cause mortality and LDL-C*

Total Deaths (%) Unadjusted Adjusted*

HR (95%CI) P value HR (95%CI) P value

Men

Q1 972 69 (7.10) 1 - 1 -

Q2 999 58 (5.81) 0.806(0.568~1.142) 0.2251 0.978(0.623~1.536) 0.9244

Q3 979 45 (4.60) 0.642(0.441~0.934) 0.0205 0.763(0.473~1.231) 0.2681

Q4 992 35 (3.53) 0.491(0.327~0.737) 0.0006 0.532(0.317~0.893) 0.0169

Q5 982 41 (4.18) 0.579(0.394~0.853) 0.0056 0.819(0.498~1.348) 0.4329

Women

Q1 1106 44 (3.98) 1 - 1 -

Q2 1094 41 (3.75) 0.951(0.622~1.456) 0.8177 1.284(0.755~2.182) 0.3561

Q3 1094 27 (2.47) 0.622(0.385~1.005) 0.0523 0.702(0.383~1.287) 0.2528

Q4 1105 35 (3.17) 0.799(0.513~1.246) 0.3222 0.912(0.514~1.619) 0.7542

Q5 1099 41 (3.73) 0.934(0.610~1.430) 0.7538 0.995(0.576~1.720) 0.9853

Participants who died during the first year were excluded.

* The multivariable models were adjusted for age, smoking, drinking, BMI, living alone, household income, educational level, rural residence, ADL disability, HDL-C, triglyceride, hemoglobin, hypertension, HBS/diabetes, history of stroke, cancer, heart disease, lung disease, liver disease, kidney disease, digestive disease, asthma, arthritis, psychological problem and memory problem.

Figure 5 Results from restricted cubic splines for the association between LDL-C and 4-year all-cause

mortality in men and women (excluding participants who died during the first year). The multivariable models were adjusted for age, smoking, drinking, BMI, living alone, household income, educational level, rural residence, ADL disability, HDL-C, triglyceride, hemoglobin, hypertension, HBS/diabetes, history of stroke, cancer, heart disease, lung disease, liver disease, kidney disease, digestive disease, asthma, arthritis, psychological problem and memory problem.

18. In the first paragraph of the Discussion section and in the abstract, it is stated that low LDL-C is associated with increased all-cause mortality risk. However, analyses are based on using low LDL-C as a reference and demonstrating a lower all-cause mortality risk in the 2 highest quintiles. It should be clarified what LDL-C values are and compared to what reference the risk differs.

Response: Thank you very much for your suggestion. The quintile analyses showed that compared with those in the first quintile of LDL-C (<83.89mg/dL), male participants in the fourth quintile of LDL-C (117.14~136.86mg/dL) had lower mortality risk, and vice versa. Further analyses using restricted cubic spline (RCS) could provide a more comprehensive spectrum for the association between LDL-C and mortality risk. The results from RCS showed that when the 20th percentile of LDL-C levels was used as the reference, lower LDL-C was associated with higher risk of 4-year all-cause mortality in men, and moderately higher LDL-C possessed lower total mortality risk, but the association was not statistically significant for very high LDL-C concentrations. Taken together, a very low plasma level of LDL-C was associated with increased all-cause mortality risk. In the revised manuscript, we have revised the statements accordingly.

References

1. Yusuf S, Bosch J, Dagenais G, et al. Cholesterol Lowering in Intermediate-Risk Persons without Cardiovascular Disease. *New England Journal Of Medicine* 2016;374(21):2021-31|.
2. Shepherd J, Cobbe SM, Ford I, et al. Prevention Of Coronary Heart-Disease with Pravastatin In Men with Hypercholesterolemia. *New England Journal Of Medicine* 1995;333(20):1301-07.
3. Ford I, Murray H, McCowan C, et al. Long-Term Safety and Efficacy of Lowering Low-Density Lipoprotein Cholesterol With Statin Therapy 20-Year Follow-Up of West of Scotland Coronary Prevention Study. *Circulation* 2016;133(11):1073-80|.
4. Zhao Y, Hu Y, Smith JP, et al. Cohort profile: the China Health and Retirement Longitudinal Study (CHARLS). *Int J Epidemiol* 2014;43(1):61-8

Reviewer 2

1. The analysis includes only those participants in the cohort who have LDL measurements at baseline, so that 34% are excluded. There should be some assessment of any potential bias that this might promote. Note that the findings are contentious so that extra care should be taken to investigate any potential bias.

Response: Thank you very much for your suggestion. We agree with your point that many individuals were excluded in the study selection procedure, which should be assessed for potential bias. We have compared the characteristics between subjects excluded and those non-excluded (Table 1). Although about half of these characteristics were significantly different ($P < 0.05$), most of the differences were not large from a clinical perspective. Moreover, we have adjusted all these potential influencing factors in multivariable analyses.

Table 1 Comparisons of characteristics between subjects excluded and those non-excluded
Characteristics Men (n=8476) Women (n=9230)

Non-excluded

(n=4981) Excluded

(n=3495) P Non-excluded

(n=5529) Excluded

(n=3701) P

Age-yr 59 (53-66) 57 (50-64) <0.0001 57 (51-65) 56 (48-64) <0.0001

BMI-kg/m² 22.40 (20.35-24.83) 22.76 (20.59-25.37) 0.0001 23.51 (21.17-26.14) 23.76 (21.29-26.52) 0.0160
 SBP-mmHg 127.67 (115.67-141.67) 128.67 (116.67-143.00) 0.0387 127.00 (114.00-143.33) 126.33 (114.67-144.00) 0.9070
 DBP-mmHg 75.33 (67.67-83.67) 76.67 (68.67-85.33) <0.0001 74.33 (67.00-82.67) 74.67 (67.67-83.67) 0.0542
 Above-average household income-no. (%) 2104 (49.68) 1737 (60.21) <0.0001 2360 (50.32) 1662 (55.31) <0.0001
 Education level-no. (%) <0.0001 <0.0001
 1 4287 (86.07) 2778 (79.58) 5134 (92.86) 3264 (88.46)
 2 592 (11.89) 542 (15.53) 347 (6.28) 350 (9.49)
 3 102 (2.05) 171 (4.90) 48 (0.87) 76 (2.06)
 ADL disability-no. (%) 716 (14.55) 480 (14.09) 0.5587 1011 (18.57) 677 (18.86) 0.7273
 Living alone-no. (%) 471 (9.46) 314 (8.99) 0.4659 832 (15.05) 631 (17.09) 0.0085
 Rural residence-no. (%) 3282 (65.89) 1842 (52.70) <0.0001 3524 (63.74) 1889 (51.04) <0.0001
 Lifestyle-no. (%)
 Smoking ever 3738 (75.24) 2470 (71.72) 0.0003 420 (7.63) 303 (8.33) 0.2209
 Drinking ever 3297 (66.43) 2228 (64.82) 0.1268 826 (15.01) 472 (12.98) 0.0067
 Disease history-no. (%)
 Hypertension 1904 (43.35) 1309 (50.54) <0.0001 2245 (45.34) 1447 (50.32) <0.0001
 Cancer 40 (0.81) 38 (1.11) 0.1646 67 (1.22) 44 (1.21) 0.9730
 Stroke 133 (2.69) 128 (3.72) 0.0074 136 (2.48) 108 (2.97) 0.1525
 Heart disease 499 (10.10) 399 (11.65) 0.0243 738 (13.47) 583 (16.10) 0.0005
 Lung disease 620 (12.54) 412 (11.98) 0.4475 508 (9.26) 293 (8.08) 0.0521
 Arthritis 1548 (31.23) 938 (27.31) 0.0001 2243 (40.80) 1256 (34.57) <0.0001
 Liver disease 197 (4.00) 154 (4.50) 0.2586 188 (3.44) 111 (3.07) 0.3353
 Kidney disease 309 (6.27) 241 (7.04) 0.1599 315 (5.75) 191 (5.28) 0.3363
 Digestive disease 1030 (20.79) 645 (18.76) 0.0221 1393 (25.37) 874 (24.06) 0.1580
 Asthma 283 (5.72) 193 (5.63) 0.8510 216 (3.93) 134 (3.69) 0.5578
 Psychological problem 60 (1.21) 53 (1.54) 0.1966 94 (1.72) 70 (1.93) 0.4498
 Memory problem 97 (1.96) 102 (2.97) 0.0029 86 (1.57) 84 (2.32) 0.0095
 Laboratory measurements
 LDL cholesterol-mg/dL 109.41 (88.92-131.06) 114.43 (93.94-140.34) 0.0024 117.91 (96.26-141.50) 116.37 (92.40-140.72) 0.1054
 Triglyceride-mg/dL 96.46 (69.03-145.14) 130.10 (85.85-213.29) <0.0001 110.63 (79.65-159.30) 124.79 (84.96-180.54) <0.0001
 HDL cholesterol-mg/dL 48.71 (39.43-59.54) 42.14 (33.63-51.61) <0.0001 50.64 (41.75-60.31) 45.62 (37.89-55.67) <0.0001
 Hemoglobin-(g/dL) 15.10 (14.00-16.20) 15.30 (14.20-16.40) 0.0077 13.60 (12.50-14.60) 13.70 (12.70-14.70) 0.0176
 ADL, activity of daily living; BMI, body mass index; SBP, systolic blood pressure; DBP, diastolic blood pressure.
 Educational level: 1, Less than lower secondary education; 2, Upper secondary & vocational training; 3, Tertiary education.

2. The wording regarding follow up is not clear to me. I assume though that there are some participants with LDL at baseline that are lost to follow up and then excluded. Again there is potential for bias and I would expect some effort to include at least a sensitivity analysis to explore the potential bias from this.

Response: We are sorry that we did not describe this clearly. During follow-up, there were 109 individuals lost to follow-up in men and 123 in women, and the follow-up rates were 97.86% and 97.82%, respectively. We have compared the characteristics between those successful followed up

and those lost to follow-up. Most of the characteristics were not statistically different between subjects followed up and those lost to follow-up (Table 2).

Table 2 Comparisons of characteristics between subjects followed up and those lost to follow-up

Characteristics	Men (n=5090)	Women (n=5652)	P
Successfully followed up	(n=4981)	Lost to follow-up	
(n=109)	Successfully followed up	(n=4981)	Lost to follow-up
(n=123)	P		
Age-yr	59 (53-66)	57 (50-63)	0.1076
BMI-kg/m ²	22.40 (20.35-24.83)	22.77 (20.78-25.57)	0.4955
SBP-mmHg	127.67 (115.67-141.67)	129.33 (118.67-150.67)	0.1870
DBP-mmHg	75.33 (67.67-83.67)	78.67 (70.00-85.00)	0.2060
Above-average household income-no. (%)	2104 (49.68)	52 (61.90)	0.0265
Education level-no. (%)	0.2778	0.0012	
1	4287 (86.07)	90 (82.57)	5134 (92.86)
2	592 (11.89)	15 (13.76)	347 (6.28)
3	102 (2.05)	4 (3.67)	48 (0.87)
ADL disability-no. (%)	716 (14.55)	15 (14.56)	0.9970
Living alone-no. (%)	471 (9.46)	11 (10.09)	0.8225
Rural residence-no. (%)	3282 (65.89)	34 (31.19)	<0.0001
Lifestyle-no. (%)			
Smoking ever	3738 (75.24)	74 (71.15)	0.3397
Drinking ever	3297 (66.34)	63 (60.58)	0.2112
Disease history-no. (%)			
Hypertension	1904 (43.35)	43 (55.84)	0.0284
Cancer	40 (0.81)	1 (0.96)	0.5758
Stroke	133 (2.69)	2 (1.90)	0.8524
Heart disease	499 (10.10)	9 (8.65)	0.6279
Lung disease	620 (12.54)	9 (8.65)	0.2352
Arthritis	1548 (31.23)	22 (21.36)	0.0321
Liver disease	197 (4.00)	2 (1.94)	0.4218
Kidney disease	309 (6.27)	11 (10.58)	0.0744
Digestive disease	1030 (20.79)	21 (20.19)	0.8816
Asthma	283 (5.72)	3 (2.94)	0.2292
Psychological problem	60 (1.21)	3 (2.88)	0.2827
Memory problem	97 (1.96)	0 (0.00)	0.2796
Laboratory measurements			
LDL cholesterol-mg/dL	109.41 (88.92-131.06)	115.21 (92.40-142.66)	0.0303
Triglyceride-mg/dL	96.46 (69.03-145.14)	103.54 (76.11-162.84)	0.1000
HDL cholesterol-mg/dL	48.71 (39.43-59.54)	47.94 (38.27-57.99)	0.5667
Hemoglobin-(g/dL)	15.10 (14.00-16.20)	15.15 (14.00-16.10)	0.7244

ADL, activity of daily living; BMI, body mass index; SBP, systolic blood pressure; DBP, diastolic blood pressure.

Educational level: 1, Less than lower secondary education; 2, Upper secondary & vocational training; 3, Tertiary education.

3. A Cox model has been used. There appears to no checking of the proportional hazards assumption. Please provide this. It may reveal that proportional hazards are not valid.

Response: Thank you for your suggestion. We have checked the proportional hazards assumptions by modeling log-log of survival curves (Figure 1). Although the results suggest that the proportional hazards assumptions were not totally valid, the extent of overlapping of survival curves for different LDL-C groups were not severe, especially among groups with large differences of survival curves. To validate the validity of our results, we performed additional logistic regression analyses on the relation between LDL-C quintiles and 4-year total mortality, which did not take into account the survival time effect. The results from logistic regression were totally in accordance with the results from Cox regression (Table 3), which assured the validity of our findings.

Figure 1 Log-log of survival curves in different LDL-C quintiles

Table 3 Logistic regression analyses on the relation between LDL-C quintiles and 4-year all-cause mortality risk

Total Deaths (%)		Unadjusted		Adjusted*	
OR (95%CI)	P value	OR (95%CI)	P value	OR (95%CI)	P value
Men					
Q1	991 88(8.88)	1	-	-	-
Q2	1008 67(6.65)	0.731	(0.525~1.017)	0.0628	0.839(0.529~1.332) 0.4569
Q3	991 57(5.75)	0.626	(0.443~0.885)	0.0079	0.730(0.455~1.171) 0.1921
Q4	1004 47(4.68)	0.504	(0.350~0.726)	0.0002	0.541(0.329~0.890) 0.0155
Q5	987 46(4.66)	0.502	(0.347~0.725)	0.0002	0.755(0.460~1.238) 0.2651
Women					
Q1	1114 52(4.66)	1	-	-	-
Q2	1102 49(4.45)	0.950	(0.637~1.417)	0.8027	1.378(0.808~2.352) 0.2394
Q3	1096 29(2.64)	0.555	(0.350~0.881)	0.0125	0.667(0.359~1.242) 0.2016
Q4	1111 41(3.69)	0.783	(0.515~1.189)	0.2504	0.933(0.525~1.659) 0.8136
Q5	1106 48(4.34)	0.927	(0.620~1.384)	0.7096	1.045(0.601~1.817) 0.8750

*Adjusted for age, smoking, drinking, BMI, living alone, household income, educational level, rural residence, ADL disability, HDL-C, triglyceride, hemoglobin, hypertension, HBS/diabetes, history of stroke, cancer, heart disease, lung disease, liver disease, kidney disease, digestive disease, asthma, arthritis, psychological problem and memory problem.

4. It is good to see that nonlinear effects of LDL are examined through use of restricted cubic splines. There are likely however to be nonlinear effects of age and there could be statistical interaction between many of the covariates.

Response: Thanks for this comment. We have checked the interactions between age and LDL-C by adding the interaction term "age*LDL-C" to the Cox model, and no interactions was shown between age and LDL-C in both men (P=0.8882) and women (P=0.1417). In addition, we have checked the interactions between LDL-C and other risk factors. The results showed that no interactions found between LDL-C and other risk factor, with the exception that the interaction between LDL-C and smoking in women was marginal significant (Table 4).

Table 4 Analyses of interactions between LDL-C and risk factors

ID	Interaction term	P value in men	P value in women
1	LDL-C*age	0.8882	0.1417
2	LDL-C*obesity	0.2972	0.3615
3	LDL-C*rural residence	0.1110	0.8224
4	LDL-C*ADL disability	0.4685	0.3462
5	LDL-C*smoking	0.5806	0.0394

- 6 LDL-C*drinking 0.9719 0.4802
- 7 LDL-C*hypertension 0.7614 0.9823
- 8 LDL-C*diabetes 0.6596 0.1295
- 9 LDL-C*heart disease 0.7774 0.8128
- 10 LDL-C*stroke 0.1523 0.2814
- 11 LDL-C*cancer 0.7232 0.8481
- 12 LDL-C*lung disease 0.6097 0.5762
- 13 LDL-C*memory disease 0.6670 0.1806
- 14 LDL-C*kidney disease 0.1601 0.1078
- 15 LDL-C*arthritis 0.3237 0.1869
- 16 LDL-C*asthma 0.3453 0.5406
- 17 LDL-C*liver disease 0.6433 0.7163
- 18 LDL-C*digestive disease 0.0965 0.2415
- 19 LDL-C*psychological disease 0.4119 0.3283

5. These have not been investigated. I would like more detail of what is meant by HBS/DM. The rates cited are high and I assume relate to diabetes and pre-diabetes taken together. This would then be consistent with other studies in China.

Response: High blood sugar (HBS)/diabetes was defined by a history of HBS/diabetes, or fasting blood glucose ≥ 6.1 mmol/L, or non-fasting blood glucose ≥ 7.8 mmol/L. As HBS/diabetes covers diabetes and pre-diabetes, the prevalence rates in men and women are relatively high. Since the self-reported question ("Have you been diagnosed with HBS/DM by a doctor?") did not distinguish HBS from diabetes, we could not define diabetes alone in this study.

6. Interpretation is based on $p < 0.05$ from, for example Table S2. There are issues of multiple testing and some qualification of this approach might be given in the study limitations.

Response: Thank you for your suggestion. We agree with your viewpoint that there were issues of multiple testing in Tables S2 and S3, which could result in Type I Error Inflation. In the revised manuscript, this has been listed as a limitation.

7. The authors acknowledge that there are issues around using all-cause mortality as the outcome. They are open to the possibility that LDL might be associated in opposite directions for different causes of mortality. This is true for other covariates also, and is a frustration of interpretation whenever a composite outcome is used.

Response: Thank you very much for your comment. Indeed, the unavailability of cause-specific mortality data prevents the analysis of different causes of mortality, which has been listed as one of the main limitations in the revised manuscript. In the future, we would perform further studies evaluating the relationship between LDL-C and cause-specific mortality when the relevant data are available.

VERSION 2 – REVIEW

REVIEWER	Shuaib Abdullah UT-Southwestern Medical Center, USA
REVIEW RETURNED	16-May-2020

GENERAL COMMENTS	The authors have modified their manuscript based on comments from the reviewers. Among the findings from the new analyses is that moderately higher LDL-C were associated with lower risk of all cause mortality in those aged 45-60 years old, as well as those > 60 years old. However, when adjusted for additional variables in the multivariable analyses, the highest quintile of LDL-C is not
--

	associated with a lower risk of all-cause mortality. 1) There is still some concern about the wording of the conclusions of the study. When the authors state that very low LDL-C is associated with higher all-cause mortality, there needs to be better clarification of the level of LDL-C at which a statistical association is present, as well as what LDL-C level this is in comparison to. Cox proportional hazards models analyses used the 1st quintile of LDL-C levels as the reference to show that moderately elevated LDL-C (117-137), but not LDL-C > 137, was associated with a lower risk of all-cause mortality in men. For spline analyses, the 20th percentile, or LDL-C of 87, was used as the reference, however it is unclear at what level of low LDL-C the increased risk is statistically significant. Adding the specific LDL-C level(s) and clarifying what the risk is in comparison to improves the clinical relevance of the findings. 2) How was activity of daily living assessed? This should be described in the methods section. 3) On page 9, lines 12-13, the authors imply that lowering high LDL-C to very low levels may be dangerous. However, studies such as FOURIER and ODYSSEY-Outcomes, in which a substantial proportion of patients in the PCSK9 inhibitor arm achieved a LDL-C < 50, have demonstrated that very low LDL-C in this population appears safe. The authors should qualify their statement to pertain to their specific results and state that very low LDL-C in populations not on lipid therapy may be associated with harm. 4) The results of the Cox analyses when those with LDL-C <50 are excluded (Table 4 in the authors response) should be added to the supplement section. 5) The lack of interactions of LDL-C with variables such as age and different disease processes should be mentioned in the Results section. 6) Are hand grip strength and albumin not available in the CHARLS cohort? 7) The median and IQR for each LDL-C quintile should be listed in a row in Tables 2 and 3.
--	---

REVIEWER	Robert M West University of Leeds, UK
REVIEW RETURNED	29-Apr-2020

GENERAL COMMENTS	I am satisfied by the revisions made by the authors.
--

VERSION 2 – AUTHOR RESPONSE

Reviewer 1

1. There is still some concern about the wording of the conclusions of the study. When the authors state that very low LDL-C is associated with higher all-cause mortality, there needs to be better clarification of the level of LDL-C at which a statistical association is present, as well as what LDL-C level this is in comparison to. Cox proportional hazards models analyses used the 1st quintile of LDL-C levels as the reference to show that moderately elevated LDL-C (117-137), but not LDL-C > 137, was associated with a lower risk of all-cause mortality in men. For spline analyses, the 20th percentile, or LDL-C of 87, was used as the reference, however it is unclear at what level of low LDL-C the increased risk is statistically significant. Adding the specific LDL-C level(s) and clarifying what the risk is in comparison to improves the clinical relevance of the findings.

Response: Thank you very much for your comments. We agree with your viewpoint that providing the

specific LDL-C level(s) possesses clinical significance. The quintile analyses showed that compared with those in the first quintile of LDL-C (≤ 84 mg/dL), male participants in the fourth quintile of LDL-C (117–137mg/dL) had lower mortality risk, and vice versa. This means that compared with the fourth quintile, the first quintile had higher mortality risk (Table 1).

Table 1 Associations between all-cause mortality and LDL-C in men

Total Deaths (%)	Unadjusted HR (95%CI)	Adjusted HR (95%CI)	Unadjusted P value	Adjusted P value
Q1 991 88(8.88)	1.928(1.353-2.747)	0.0003 1.652(1.040-2.623)	0.0003	0.0335
Q2 1008 67(6.65)	1.413(0.973~2.051)	0.0693 1.351(0.835~2.185)	0.0693	0.2200
Q3 991 57(5.75)	1.232(0.837~1.812)	0.2904 1.292(0.799~2.091)	0.2904	0.2966
Q4 1004 47(4.68)	1 - 1 -			
Q5 987 46(4.66)	0.987(0.657~1.482)	0.9506 1.327(0.799~2.201)	0.9506	0.2741

*The multivariable models were adjusted for age, smoking, drinking, BMI, marital status, household income, educational level, rural residence, ADL disability, hand grip strength, HDL-C, triglyceride, hemoglobin, hypertension, HBS/diabetes, history of stroke, cancer, heart disease, lung disease, liver disease, kidney disease, digestive disease, asthma, arthritis, psychological problem and memory problem.

The results from restricted cubic spline (RCS) showed that when the 20th percentile of LDL-C levels (84mg/dL) was used as the reference, lower LDL-C (< 84 mg/dL) was associated with higher risk of 4-year all-cause mortality in men, and moderately higher LDL-C (84-135mg/dL) possessed lower total mortality risk, but the association was not statistically significant for much higher LDL-C concentrations (> 135 mg/dL). Therefore, the results from RCS were generally in accordance with those from quintile analyses. Taken together, compared with moderately elevated LDL-C (e.g., 117-137mg/dL), a lower plasma level of LDL-C (e.g., ≤ 84 mg/dL) was associated with an increased risk of 4-year all-cause mortality in middle-aged and elderly Chinese men. However, we have to admit that we cannot formulate the exact LDL-C level(s) threshold for mortality risk in this preliminary study, which call for rigorous large-scale population studies in the future.

Therefore, we made corresponding revisions about the wording of the conclusions as follows:

Compared with moderately elevated LDL-C (e.g., 117-137mg/dL), a lower plasma level of LDL-C (e.g., ≤ 84 mg/dL) was associated with an increased risk of 4-year all-cause mortality in middle-aged and elderly Chinese men.

In addition, we have added the statements about LDL-C level(s) threshold in the discussion section as follows: At last, well-designed, large-scale population studies are needed to formulate the specific LDL-C level(s) threshold for mortality risk in the future.

2. How was activity of daily living accessed? This should be described in the methods section.

Response: Activity of daily living (ADL) covers the following items: dressing, bathing and showering, eating, getting in/out bed, using the toilet, and controlling urination or defecation. Every item in the ADL scale has a four-scale answer for each question: “no difficulty”, “have difficulty but can still do it”, “have difficulty and need help”, and “can not do it”. ADL was assigned a value of 0 if the respondents had no difficulty in all these activities and 1 otherwise. In the revised manuscript, we have added these descriptions in the method section.

3. On page 9, lines 12-13, the authors imply that lowering high LDL-C to very low levels may be dangerous. However, studies such as FOURIER and ODYSSEY-Outcomes, in which a substantial proportion of patients in the PCSK9 inhibitor arm achieved a LDL-C < 50 , have demonstrated that very low LDL-C in this population appears safe. The authors should qualify their statement to pertain to their specific results and state that very low LDL-C in populations not on lipid therapy may be associated with harm.

Response: Thank you for your suggestion. In the revised manuscript, we have revised these statements according to your suggestion.

4. The results of the Cox analyses when those with LDL-C <50 are excluded (Table 4 in the authors response) should be added to the supplement section.

Response: The results of the Cox analyses when those with LDL-C <50 are excluded have been added to the supplementary materials of the revised manuscript.

5. The lack of interactions of LDL-C with variables such as age and different disease processes should be mentioned in the Results section.

Response: Thank you for your suggestion. In the revised manuscript, we have added the results of interactions of LDL-C with potential risk factors of mortality to the supplementary materials. In addition, we have mentioned interactions of LDL-C with potential risk factors of mortality in the Results section as follows: No interactions were found between LDL-C and potential risk factors of mortality, with the exception that the interaction between LDL-C and smoking in women was statistically significant (P=0.0498, Supplementary Table S2).

6. Are hand grip strength and albumin not available in the CHARLS cohort?

Response: We have rechecked the CHARLS database, and found that hand grip strength was available, but albumin was not available. Hand grip strength was measured with a dynamometer (Yuejian™ WL-1000, Nantong, China) in kilograms (kg) twice on each hand. In the revised manuscript, hand grip strength was additionally adjusted in the multivariable model. The mean score of two measures in the dominant hand was used in the analysis. The results showed that the association in men persisted (Figure 1 and Table 2). In the revised manuscript, we have added descriptions about hand grip strength in the methods section, and made corresponding revisions about the analysis results.

Table 2 Associations between all-cause mortality and LDL-C

Total Deaths (%)	Unadjusted	Adjusted*
HR (95%CI)	P value	HR (95%CI) P value
Men		
Q1 991 88(8.88)	1 - 1 -	
Q2 1008 67(6.65)	0.733(0.533~1.007)	0.0554 0.818(0.531~1.260) 0.3619
Q3 991 57(5.75)	0.639(0.458~0.892)	0.0084 0.782(0.507~1.208) 0.2677
Q4 1004 47(4.68)	0.519(0.364~0.739)	0.0003 0.605(0.381~0.962) 0.0335
Q5 987 46(4.66)	0.512(0.359~0.732)	0.0002 0.803(0.506~1.274) 0.3520
Women		
Q1 1114 52(4.67)	1 - 1 -	
Q2 1102 49(4.45)	0.960(0.650~1.419)	0.8394 1.245(0.749~2.071) 0.3985
Q3 1096 29(2.65)	0.565(0.359~0.890)	0.0138 0.626(0.345~1.136) 0.1233
Q4 1111 41(3.69)	0.792(0.526~1.192)	0.2632 0.852(0.489~1.483) 0.5704
Q5 1106 48(4.34)	0.926(0.625~1.371)	0.7007 0.958(0.563~1.630) 0.8736

*Adjusted for age, smoking, drinking, BMI, marital status, household income, educational level, rural residence, ADL disability, hand grip strength, HDL-C, triglyceride, hemoglobin, hypertension, HBS/diabetes, history of stroke, cancer, heart disease, lung disease, liver disease, kidney disease, digestive disease, asthma, arthritis, psychological problem and memory problem.

Figure 2 Results from restricted cubic splines for the association between LDL-C and 4-year all-cause mortality in men and women, respectively. The multivariable models were adjusted for age, smoking, drinking, BMI, marital status, household income, educational level, rural residence, ADL disability, hand grip strength, HDL-C, triglyceride, hemoglobin, hypertension, HBS/diabetes, history of stroke, cancer, heart disease, lung disease, liver disease, kidney disease, digestive disease, asthma, arthritis, psychological problem and memory problem.

7. The median and IQR for each LDL-C quintile should be listed in a row in Tables 2 and 3.

Response: Thank you for your suggestion. In the revised manuscript, we have listed the median and IQR for each LDL-C quintile in the last rows of Tables 2 and 3.

VERSION 3 – REVIEW

REVIEWER	Robert M West University of Leeds UK
REVIEW RETURNED	19-Jun-2020
GENERAL COMMENTS	The authors have addressed all of my concerns appropriately.